



# Detecting the occurrence of preferential flow in soils with stable water isotopes

Jonas Pyschik[1] and Markus Weiler[1]

[1]Chair of Hydrology, University of Freiburg, Freiburg, Germany

**Correspondence:** Jonas Pyschik (science@jonas-pyschik.com)

**Abstract.** Subsurface flow in preferential pathways in soils may transport water more rapidly than the soil matrix, which may be quickly activated during precipitation events and enhancing infiltration or interflow. Vertical pathways are particularly important for runoff generation. However, identifying these pathways is challenging because traditional methods such as piezometers, soil moisture sensors, or hillslope trenches do not adequately capture the spatial scale and frequency of preferential flow features, while other experimental techniques like dye tracing are labor-intensive and invasive. In this study, we introduce a novel method to identify the locations of preferential flow by analysing vertical soil profiles of stable water isotope. Across four catchments, we drilled 100 soil cores (1-3 m deep) per catchment and analyzed the stable isotope composition of the soil water in 10-20 cm depth intervals to construct depth profiles. We employed clustering techniques to group soil-water isotope profiles and selecting those that match to a seasonal sampling date to establish a reference profile for each catchment using LOESS regression, representing profiles influenced solely by matrix infiltration. Deviations from these reference profiles were then used as indicators of being influenced by vertical or lateral preferential flow. Our results revealed evidence of preferential flow in all studied catchments. Especially in the alpine catchment with highly heterogeneous soils many profiles showed distinct preferential flow features, including multiple, vertically independent pathways occurring at variable depths, even among adjacent profiles. These findings demonstrate the feasibility of using soil water isotope profiles to assess preferential flow pathways highlighting the substantial spatial and vertical variability of preferential flowpaths at hillslope and catchment scale.

## 1   Introduction

Preferential flow (PF) is an important process in soils, facilitating the rapid transport of substantial amounts of water through the soil (Anderson et al., 2009; Beven and Germann, 2013; Angermann et al., 2017). During preferential flow, water rather flows through preferred pathways than the surrounding soil matrix, bypassing much of the soil (Kung, 1990; Nimmo, 2021). Preferential flow does not occur during every rainfall or infiltration event, but when it is activated, it can account for a significant proportion of annual flow though the soil, ranging from 16-27% (Eguchi and Hasegawa, 2008). Once formed, these pathways can persist across multiple events, with soil saturation playing a key role in determining which flow features become active (Wessolek et al., 2009; Anderson et al., 2009; Guo et al., 2014).



Understanding the occurrence of preferential flow is important for two reasons: understanding and predicting runoff gen-
eration and contaminant transport. Preferential flow pathways can transport water much faster than the soil matrix(Anderson
et al., 2009; Beven and Germann, 2013; Angermann et al., 2017), with reported lateral velocities ranging from 0.3 m/s up to
0.8 m/s (Anderson et al., 2009; Wilson et al., 2016). These high velocities can lead to rapid responses in streamflow. Addi-
tionally, the complex and interconnected nature of preferential flowpaths allows water to move through the soil with minimal
interaction with the soil matrix (Anderson et al., 2009; Angermann et al., 2017). This reduces a soil's capacity to filter and
retain contaminants, potentially leading to the rapid transport of pollutants into the groundwater or rivers (Khan et al., 2016).

Three primary modes of preferential flow can occur, also simultaneously, in soils: fingered flow, funneled flow and macropore
flow (Kodešová et al., 2012; Nimmo, 2021).

*Fingered flow* occurs when the infiltration wetting front reaches a more permeable soil layer and instabilities cause down-
wards flow in "fingers" (Selker et al., 1992a; Nimmo, 2021). It is driven by gravity with the highest water content at the finger
tip, widening through diffusion into the soil matrix (Nimmo, 2021). Flow velocity within a finger decreases over time, thus
more fingers form to transport an equal amount of water (Selker et al., 1992b).

*Funneled flow* occurs when infiltrating water reaches an impermeable or less conductive soil-layer which then causes the
flow to be channeled laterally (Kung, 1990; Kotikian et al., 2019; Heilig et al., 2003; Nimmo, 2021). This funneling effect can
guide water into vertical preferential flowpaths like macropores (Heilig et al., 2003).

*Macropore flow* is the predominant preferential flow process in many soils, transporting up to 95 % of water through less
than 0.3 % of the total soil volume (Watson and Luxmoore, 1986). This type of flow includes various forms such as biopore
flow, inter-aggregate flow, and preferential flow within soil aggregates (Bouma et al., 1977; Luo et al., 2008; Kodešová et al.,
2012). Macropores are created through multiple mechanisms, including biological activity from soil fauna (e.g. earthworms,
gophers, beetles) (Weiler and Naef, 2003; Badorreck et al., 2012)), plant roots (Leslie and Heinse, 2013; Pinos et al., 2023)),
soil cracks caused by shrinkage (Thomas et al., 2013; Demand et al., 2019a)) and soil pipes fromed by internal erosion (Jones,
2010; Glenn V. Wilson et al., 2013)). These macropores can persist for years, and in some cases, they may remain active for
decades (Beven and Germann, 1982). Within a soil profile, smaller diameter flowpaths can restrict preferential flow throughout
the whole soil (Paradelo et al., 2016). However, even small continuous macropores can transport significant volumes of water
if they maintain connectivity (Bouma, 1981). Macropore flow is initiated at or close to topsoil saturation (Weiler and Naef,
2003). During water flow in macropores, they are typically not filled entirely with water but a water film forms along the walls
if the pore, leaving the central part of the pore empty (Bouma et al., 1977; Bouma and Dekker, 1978). The complex network
of macropores, particularly soilpipes in hillslopes, can extent up to 190 m and can contribute up to 50 % to overall discharge
in streams (Jones, 2010; Wilson et al., 2016). This extensive and efficient transport network underscores the importance of
macropore flow in both hydrological and contaminant transport processes within soils.

In hillslopes the interplay of funneled and macropore flow is also called subsuface stormflow (SSF) (Guo et al., 2014).
Macropores can rapidly transport infiltrating water to depths where it laterally discharges through soilpipes and funneled flow
(Kelln et al., 2007). While SSF only occurs in approximately 1/3 of rain-events, it may transport up to 90 % of rainfall into the





soil and to the stream (Beasley, 1976). Observed SSF velocities in preferential flowpaths are higher than in the surrounding soil matrix (Angermann et al., 2017).

Detecting preferential flow is challenging due to the inherent complexity and heterogeneity of the relevant soil structures. Preferential flow occurs in a complex network of earthworm channels, inter-aggregate space (Bouma and Dekker, 1978) and root channels meso/micropores (Luo et al., 2008). However, simply identifying these voids is not sufficient, as not all voids carry water during preferential flow (Beven and Germann, 1982; Nimmo, 2021). Also, field surveys are only a snapshot of an environmental system, but flowpath arrangements can change completely within one year (Wessolek et al., 2009; Beven, 2019). Additionally, even when preferential flow features are successfully identified at the plot scale, these findings can not be easily extrapolated to the whole catchment. This limitation arises because hillslopes exhibit varying hydrological responses influenced by micro-topography, soil properties, and curvature, leading to significant spatial variability in preferential flow pathways. (Beven and Germann, 1982; Woods and Rowe, 1996; Gazis and Feng, 2004; Guo et al., 2014; Demand et al., 2019a). As a result, understanding and predicting preferential flow at broader scales remains a complex challenge that requires integrating spatial and temporal variability in soil structure and hydrological processes.

Many different methods exist to identify and partly quantify preferential flow in soils, but each has its own limitations:

- *Dye tracers and sprinkling experiments* are commonly used to visualize flowpaths in soils but have the disadvantage of destroying the site during excavation for surveying (Weiler and Naef, 2003; Alaoui and Helbling, 2006). Additionally, these techniques may miss flowpaths that transport pre-event water, as these pathways will not be marked by the dye of the event water (Beven and Germann, 2013).

- *Soil moisture sensors* located in a depth profile can detect preferential flow, by detecting rapid increases in moisture content that exceed what would be expected from matrix flow alone, or by observing non-sequential responses in sensors placed at different depths (Blume et al., 2009; Demand et al., 2019a). *Lysimeters* can be used in a similar way as the soil moisture sensors, if the outflow response occurs faster than predicted by advective or dispersive matrix transport (Małoszewski et al., 2006). Both methods, however, are limited by their spatial scale and require significant installation efforts.

- *Ground penetrating radar (GPR)* can be used to detect soilpipes by identifying the pipe walls and tracking them spatially through the hillslope. The main drawback of this technique is its limited resolution, which restricts its ability to detect only larger flow features (Holden et al., 2002)

- *X-Ray and computer tomography (CT)* can be used to visualize macropore networks or other pore structures in soils (Katuwal et al., 2015; Sammartino et al., 2015; Paradelo et al., 2016). However, as the sample volumes of the CTs are related to the resolution, the undisturbed soil cores are difficult to obtain and, due to edge effects induced by the soil coring, can alter the measured flow.

- *Trenched hillslopes* have also been used to detect preferential flow, particularly lateral flow. Variable wetness or observed flow volumes on the hillslope face (the excavated cross section of hillslope soil) is the result from different flowpaths



(Woods and Rowe, 1996; Tromp-van Meerveld and McDonnell, 2006; Scaini et al., 2017). However, trench excavation and monitoring is very labor intensive and results can only be interpreted at the plot scale.

This study aims to detect lateral and vertical preferential flowpaths in soils using stable water isotopes. Stable water isotopes are controlled by several factors (Sprenger et al., 2016b). The most relevant factor, fractionation, results from the phase transitions of water from the liquid to the gaseous phase and vice versa. Water vapor, which contains the heavier isotopes $^2$H and $^{18}$O, has a higher vapor pressure than water vapor of the lighter isotopes $^1$H and $^{16}$O , leading to a slower diffusion rate of heavy water molecules(Leibundgut et al., 2009). This results in a preference for light water molecules in the during evaporation and for heavier isotopes during condensation (Leibundgut et al., 2009). The fractionation effect is temperature-dependent, with colder conditions enhancing the enrichment of heavy isotopes in the liquid phase and lighter isotopes in the vapor phase (Dansgaard, 1964).

Stable isotope signatures in precipitation vary spatially due to factors such as the *continental-effect*, where precipitation is progressively depleted of heavy molecules with increasing distance from the ocean. By favouring heavy water molecules in condensation, the air masses gradually loose heavy isotopes (Leibundgut et al., 2009; Liu et al., 2010). Conversely, high-intensity precipitation events can lead to *rain-out* fractionation, which favours lighter isotopes (Araguás-Araguás et al., 2000). *Altitude* also plays a role, as cooler temperature at higher elevations enhance condensation, leading to a depletion of heavier molecules (Ambach et al., 1968; Araguás-Araguás et al., 2000). In the Alps, this effect causes a change of -0.2 ‰ per 100 m elevation for $\delta^{18}O$ (Ambach et al., 1968).

Seasonal variations in temperature also influence isotope ratios, especially in temperate regions. During summer, precipitation tends to have more negative isotope signatures compared to winter (Rozanski et al., 2013). In particular the seasonal variation and rain-out effects provide a dynamic input of isotopic signatures into soils, making them ideal tracer to determine water origin and flow processes within the soil (Gazis and Feng, 2004; Garvelmann et al., 2012; Mueller et al., 2014; David et al., 2018). Different flow mechanisms within the soil modify these signatures, creating distinct patterns in isotope depth profiles. By analyzing the isotopic signature of precipitation and soil cores, it is possible to infer the flow mechanisms operating within the soil (Gazis and Feng, 2004; Stumpp and Hendry, 2012).

During a rain event, precipitation first infiltrates into the topsoil, displacing older stored water and altering the isotopic signature with that of the recent precipitation (Zhao et al., 2013). The infiltrating water causes vertical transport (translatory flow) of the event water propagating through the soil matrix (Gazis and Feng, 2004; Mueller et al., 2014). This new event water pushes the older pre-event water deeper into the soil (Scaini et al., 2017). This process preserves the seasonal isotopic signature of precipitation within the soil, for example for observations during summer with higher isotopic ratios from recent summer events near the surface and lower ratios from older winter precipitation at greater depths (Gehrels et al., 1998; Garvelmann et al., 2012; Eisele, 2013; Sprenger et al., 2016a).

However, other processes can modify the isotopic seasonality within the soil. Evaporation causes an enrichment of the heavier isotopes in the topsoil (Gazis and Feng, 2004; Garvelmann et al., 2012; David et al., 2018). Advective and dispersive transport further dampens the seasonal isotope signal with increasing depth due to stronger mixing with preexisting event water (Thomas et al., 2013). This mixing can also occur due to rising and falling groundwater levels (Uchida et al., 2005; Garvelmann





et al., 2012). In hillslopes, uplslope soils preserve the precipitation signal, but further downslope, lateral flow in the soil leads to mixing and a damped isotopic signal. At the footslope, the seasonality signal may be entirely lost (Garvelmann et al., 2012).

Preferential flow also distinctly alters the seasonality signal (Mathieu and Bariac, 1996; Thomas et al., 2013; Cheng et al., 2014). With preferential flow young event water can bypass older, stationary water in the soil matrix, and reaches deeper layers, thereby modifiying the isotopic profiles locally (Gazis and Feng, 2004; Peralta-Tapia et al., 2015). Initially, water in preferential flow paths reflects the isotopic signature of the recent event, Over time, however, the signatures shifts towards those of pre-event water (Leaney et al., 1993; Gehrels et al., 1998; Kelln et al., 2007) due to lateral infiltration and exchange processes between the preferential pathways and the surrounding matrix (Weiler and Naef, 2003; Angermann et al., 2017). This exchange causes mixing, resulting in a composite isotopic signature (Leaney et al., 1993).

To assess the occurrence of preferential flow at the catchment scale and its relation to landuse and topography, we focus on identifying isotopic alterations in the soil matrix that indicate the occurrence of preferential flow. Previous studies have derived these alterations by comparing isotope profiles either to modeling results of soils without preferential flow (Mathieu and Bariac, 1996), or by analyzing soil cores sampled at different times (Cheng et al., 2014). In contrast, we compare simultaneous profiles against "reference profiles" derived from a subset of the data to identify deviations indicative of preferential flow.

## 2 Methods

### 2.1 Study Sites

Our research was conducted in four first-order catchments in three distinct low mountain ranges (Sauerland, Ore Mountains and Black Forest) and alpine landscapes (Tyrolean Alps). These sites were selected to capture a range of topographical, climatic, and land use conditions relevant to the study of preferential flow and isotopic signatures in soils.

The Obere Brachtpe (313–514 m a.s.l.) in the Rheinish Massif (*Sauerland*) covers a drainage area of approximately $47.0 \text{ km}^2$ (Gauge Husten). The dominant land uses are spruce forests and pastures. The mean annual temperature is 9.1°C, and the mean annual precipitation is 1227 mm, with 15-20% of the precipitation falling as snow. The main soil types are loamy Cambisols, Leptosols, and Stagnosols (Chifflard et al., 2008). The isotopic precipitation seasonality varies between $-40.7 \pm 7.0‰$ for $\delta^2 H$ and $-5.8 \pm 0.9‰$ for $\delta^{18}O$ in summer, and $-68.0 \pm 11.7‰$ for $\delta^2 H$ and $-9.5 \pm 1.5‰$ in winter (Nelson et al., 2021). All isotope values in this paper are expressed relative to Vienna Standard Mean Ocean Water (VSMOW).

Rütlibach and Eberbach (340–585 m a.s.l.), located at the western edge of the *Black Forest*, Germany, are characterized by patches of grasslands and coniferous and broadleaf forests. The area receives 970 mm of precipitation annually, and the mean annual temperature is 11°C. The predominant soil types are Cambisols on periglacial drift covers, with Gleysols and Colluviosols near the streams (Bachmair and Weiler, 2012, 2014). The isotopic precipitation seasonality varies between $-41.8 \pm 6.6‰$ for $\delta^2 H$ and $-6.2 \pm 0.9‰$ for $\delta^{18}O$ in summer, and $-79.9 \pm 12.5‰$ for $\delta^2 H$ and $-10.7 \pm 1.3‰$ in winter (Nelson et al., 2021).

The Padasterbach catchment (elevation 1060 - 2602 m a.s.l.) in the *Tyrolean Alps*, Austria, has the highest elevations among the research catchments. The landscape is predominantly spruce forest in the valleys and pastures with rock cliffs on the slopes.





**Figure 1.** The four research catchments Sauerland (51.3548° N 7.9836° E, Germany), Black Forest (47.9262° N 7.8268° E, Germany), Alps (47.0023° N 11.4312° E, Austria) and Ore Mountains (50.6987° N 13.2304° E, Germany)

In 2024, the mean annual temperature was 6.1 °C with 1280 mm of annual precipitation. Summer precipitation signatures are $-53.6 \pm 9.7$‰ for $\delta^2 H$ and $-8.3 \pm 1.1$‰ for $\delta^{18}O$, and winter precipitation signatures are $-112.9 \pm 14.0$‰ for $\delta^2 H$ and $-15.2 \pm 1.7$‰ (Nelson et al., 2021).

The Freiberger Mulde in the *Ore Mountains*, Germany (elevation 446 - 850 m a.s.l.), has a mean annual temperature of 6.6°C and a mean annual precipitation of 930 mm. The land use is mostly spruce forest, pastures, and cropland. The predominant soil type is Stagno-gleyic Cambisols (Heller and Kleber, 2016). Summer precipitation signatures are $-46.6 \pm 6.6$‰ for $\delta^2 H$





and $-6.8 \pm 0.8‰$ for $\delta^{18}O$, and winter precipitation signatures are $-84.9 \pm 13.1‰$ for $\delta^2 H$ and $-11.5 \pm 1.5‰$ (Nelson et al., 2021).

## 2.2 Soil Water Isotope Sampling and Analysis

Soil sampling was conducted in each catchment using an electrical auger (Makita HM1810) to drill 100 holes to refusal depth.
After extracting the core from the soil, the core was shaded and protected from rain to prevent isotopic alteration during sampling (Garvelmann et al., 2012). To avoid cross-contamination from core extraction or reinsertion, the top 5 cm of each core was scraped off and discarded. Soil samples were collected in consecutive depths intervals (10-20 cm), with finer sampling resolution closer to the soil surface to capture detailed vertical isotopic profiles.

Samples were stored into aluminum-laminated plastic bags (WEBER Packing GmbH; CB400-420BRZ; 500 ml) and sealed
with a ziplock (Gralher et al., 2021). The samples were refrigerated and analyzed within four weeks to minimize isotopic shifts that could occur over time (Gralher et al., 2021). For isotopic analysis, the samples were processed using the direct vapor equillibration method (Wassenaar et al., 2008; Gralher et al., 2021). This involved inflating the sample bags and identical calibration standard bags with dry air before heat-sealing them. A silicone blot was then added to each bag as a septum. The samples were stored under constant climatic conditions (20°C ± 1°C) for 48 hours to ensure isotopic equilibrium between the
liquid and vapor phases within the bags.

After equilibration, the headspace vapor from the bags was analyzed using a laser spectrometer (Picarro LX-i2130 and LX-i2120). A cannula connected to the analyzers inlet port was inserted through the septum of each bag for sampling. Calibration standards were co-measured after every 15 samples. Most samples were measured by VapAuSa with identical stability criteria as reported in the technical note (Pyschik et al., 2025a). All measurements were referenced to the VSMOW-SLAP scale using
in-house calibration standards, and isotope signatures were reported in the standard delta notation, representing the per-mil difference relative to Vienna Standard Mean Ocean Water (VSMOW) (Craig, 1961).

## 2.3 Reference Profiles and Identification of Preferential Flow

The identification of preferential flow in stable water isotope profiles relies on comparing measured values to established "reference profiles". The reference profiles aim to define a typical isotope profile under the premise of solely vertical matrix
flow, thereby maintaining the isotopic seasonality of precipitation while accounting for a degree of dampening resulting from mixing processes and dispersion in the soil. To obtain these profiles, the entire dataset was subdivided by catchment and sampling period groups, hereafter referred to as subsets. For each subset the $\delta^{18}O$ measurements were scaled to the range of $\delta^2 H$ using a simple linear regression, yielding $scaled^{18}O$ values. This transformation was performed to consolidate the information into a single range of values, thereby increasing the number of effective support points for subsequent polynomial
fitting. A 3rd-degree polynomial regression was fitted to each soil profile using the R stats package R Core Team (2024) and its poly function (figure 2.1). The three coefficients and the intercept for each polynomial were then clustered using the k-means algorithm from the R caret package Kuhn (2008). The optimal number of clusters, consistently found to be three, was determined using the silhouette method of the caret package. The cluster which most accurately represented the anticipated



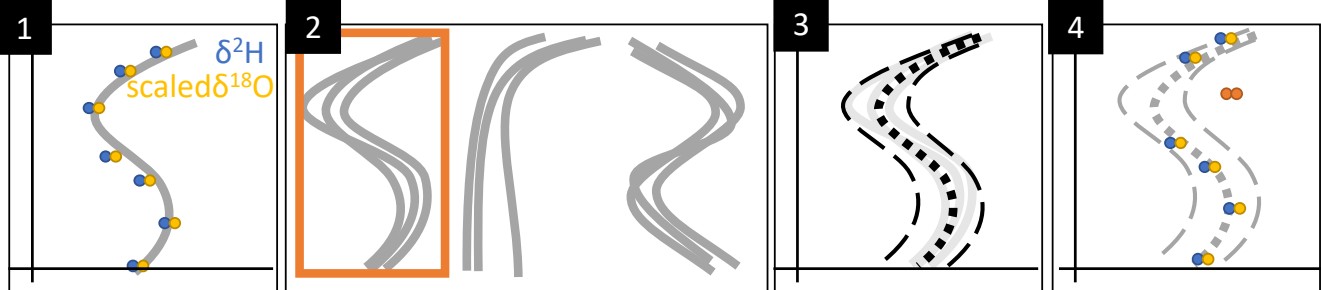

**Figure 2.** Visualization of the methodology: (1) scale $\delta^{18}$O to $\delta^2$H value range and fit a polynomial function, (2) cluster the fitted functions of all profiles and select the profile cluster which matches the isotopic seasonality, (3) calculate the mean and two standard deviation for the cluster to define a reference profile, (4) reference each individual profile and identify outliers which correspond to preferential flow

isotopic seasonality profile (Gehrels et al., 1998; Eisele, 2013; Sprenger et al., 2016a) was then visually selected to be used as the basis for the reference profile of the subset (figure 2.2). For example, if a subset was sampled in late summer, the cluster with high $\delta$-value summer precipitation ratios close to the soil surface and with low $\delta$-value winter precipitation ratios further down in the profile, was selected. The polynomial profiles within the selected cluster were then used to calculate the reference profile for the whole subset. This was achieved by calculating a LOESS regression of the selected polynomial profiles and adding a $\pm$ two standard deviation (figure 2.3). Measured values which lie within this range are considered to originate from vertical matrix flow. Only depths where both $\delta^2$H and $scaled^{18}$O values exceeded the reference range were considered for potential preferential flow (figure 2.4). These values were then compared to the value range of the topsoil (10-20 cm depth), under the assumption that preferential flow would exhibit an isotopic signature similar to that of the topsoil, as it most accurately reflects the current seasonal precipitation input. Only when both $\delta^2$H and $scaled^{18}$O fell within the range of the topsoil and reference, these depths of the divergences from the reference profile were considered to be influenced by preferential flow.

## 2.4 Mixing Models

Additionally, the profiles and depths classified as being influenced by preferential flow were subjected to a mixing analysis to determine the proportions of event water. A two end-member mixing model was applied using the R *simmr* package (Govan and Parnell, 2023). The inputs for the model included two sources: (1) the value range of the isotope signature of the topsoil at a depth of 10-20 cm of the reference and (2) the value range at the depth of the divergence of the reference profile. The mixture was charaterized by the actually measured (or scaled) $\delta^2$H and $\delta^{18}$O values of the divergence.

$$\delta_{\text{mixture}} = f \cdot \delta_{\text{topsoil}} + (1 - f) \cdot \delta_{\text{reference}} \tag{1}$$



where $\delta_{\text{mixture}}$ is the measured isotope value at a given depth, $\delta_{\text{topsoil}}$ is the isotope value at 10-20 cm depth, $\delta_{\text{reference}}$ is the reference isotope value at the same depth as the mixture, and $f$ is the proportion of the topsoil contribution to the mixture. By solving this equation, the proportion of preferential flow ($f$) at the identified depths was determined.

## 2.5 Spatial analysis

To analyse the spatial and topographic factors influencing preferential flow, all soil profiles were georeferenced. Elevation, slope and aspect were derived from Digital Elevation Models (DEM, European Space Agency (2024), 30 m resolution) for each sampling point. In addition, satellite-based land use data from the European Space Agency (ESA) were incorporated to investigate correlations between terrain characteristics and preferential flow paths (Zanaga et al., 2021). These variables were incorporated into the dataset to identify potential predictive patterns.

Logistic regression analysis was used to assess the influence of catchment, landuse, slope, elevation and aspect on the occurrence of preferential flow. Aspect, originally represented as a linear variable in degrees, was transformed into circular coordinates to account for its directional nature. The cosine transformation was applied, resulting in the following equations:

$$\text{Aspect}_{\text{cos}} = \cos\left(\frac{\text{Aspect} \cdot \pi}{180}\right) \tag{2}$$

These transformations made it possible to distinguish between north- or south-facing slopes ($\text{Aspect}_{\text{cos}}$). By that, north facing Aspect values (0-45° and 315-360°) are therefore represented by $\text{Aspect}_{\text{cos}}$ values close to 1.

## 3 Results

### 3.1 Reference profiles

The derived reference profiles (Figure 3) show large differences in isotopic signature variations as well as profile shapes among the catchments as well as between sampling seasons within each catchment.

Ore Mountains (OM) were sampled in mid summer (OM-summer; July 2022; 59 profiles) and end of winter (OM-winter; March 2023; 28 profiles). The summer profile shows a uniform pattern with only the topsoil diverging to more positive $\delta$-values. This deviation is expected from typical values of isotopic signatures in summer precipitation, as well as evaporation induced fractionation. With $\delta^2$H and $scaled^{18}$O signatures around -50 ‰ in the topsoil, the profile shows lower $\delta^2$H and $scaled^{18}$O signatures with depth, reaching -75 ‰ in 2 m depth. This water probably originated from the past winter. The summer profile has one of the highest standard deviations ($\delta^2$H and $scaled^{18}$O = ± 25 ‰). In contrast, the end-of-winter profile portrays a more defined seasonality shape with low $\delta$-values near the surface ($\delta^2$H and $scaled^{18}$O = -80 ‰) gradually increasing to higher $\delta^2$H and $scaled^{18}$O values of -65 ‰ in 2 m depth. Here, the seasonality is visible: low $\delta$-value winter precipitation replaces the topsoil water, causing higher $\delta$-value summer water to propagate downwards.

In the Black Forest (BF) catchment, samples were collected in spring (BF-spring; May 2022; 65 profiles) and summer (BF.summer; August 2022; 26 profiles). Here the spring profile shows a smooth seasonal gradient with higher $\delta$-value from





spring water near the surface, low $\delta$-value of winter water ($\delta^2$H and $scaled^{18}$O = -60 ‰) in 1 m depth, and a slight increase in $\delta$-value near 2 m, originating from past fall ($\delta^2$H and $scaled^{18}$O = -55 ‰). In comparison, the summer profile has higher $\delta^2$H and $scaled^{18}$O values with summer signatures of -30 ‰ near the surface and increasingly lower values at 1.7 m. These values are similar to the $\delta$-values ($\delta^2$H and $scaled^{18}$O = -60 ‰) of the winter peak in the spring profile. The topsoil water from the spring profile has propagated downwards to 0.5 m in the summer profile.

Sauerland (SL) samples were taken in summer (SL-summer; June/July 2022; 64 profiles) and fall (SL-fall; October 2022; 49 profiles). Both reference profiles show variations with depth, indicating the preservation of isotopic seasonality in precipitation. The summer profile shows higher $\delta^2$H and $scaled^{18}$O value summer precipitation at the surface (-35 ‰), decreasing with depth to the winter signatures ($\delta^2$H and $scaled^{18}$O = -55 ‰) in 0.5 m. A slight $\delta$-value increase is visible in 1.2 m, decreasing again to winter signatures ($\delta^2$H and $scaled^{18}$O = -55 ‰) in 2 m depth. The fall profile shows a more uniform shape with a low standard deviation of $\pm$ 2.5 ‰ .

The Tyrolean Alps (TA) catchment, sampled only once in fall (TA-fall); September 2022; 102 profiles), shows the largest seasonality and variability. The large standard deviation ($\pm$ 25 ‰) originates from the altitude effect (Ambach et al., 1968) due to a large (700 meter) elevation gradient within the catchment. Topsoil signatures indicate summer precipitation ($\delta^2$H and $scaled^{18}$O = - 50 ‰), decreasing to winter signatures ($\delta^2$H and $scaled^{18}$O = -90 ‰) in 1 m depth. Below, $\delta^2$H and $scaled^{18}$O ratios increase slightly to -80 ‰ .

## 3.2 Identified preferential flow

By applying the reference profile thresholds to each individual profile, we identified 63 profiles from the total of 393 that met the preferential flow criteria (Table 1). Most profiles exhibit a shape akin to that depicted in figure 4-left, with all measurements laying within the reference profile interval. Figure 4-right shows a example profile with multiple identified preferential flowpaths. Most of the profiles identified for preferential flow (42) were located in the alpine catchment, the remaining equally distributed among the other three catchments. In the Alps, profiles often had multiple depths in which preferential flow was identified (mean: 2.32 depths in each identified profile). Preferential flow was predominantly identified at a depth of 50 - 65 cm, with a decrease observed at greater depths (Fig. 5). But also below 1 m preferential flow was frequently identified in the profiles.

The identified preferential flow could originate from vertical or lateral pathways. If preferential flow is identified at only one depth, it is impossible to determine whether the flow was lateral or vertical. However, in profiles where multiple depths have identical preferential flow isotopic signatures, this suggests that the water has moved vertically through the profile, like Profile TA-fall 7 shown in Figure 4. Additionally, these samples were taken in close proximity along a transect, with samples 9-11 positioned farthest upslope, samples 5-8 in the middle, and sample 4 situated further downslope. In these profiles, preferential flow was identified at similar depths of approximately 50 cm and 110-130 cm. This pattern suggests the presence of a lateral preferential flow pathway in a certain depth range, such as a layer boundary with a transition to less conductive material, which forces infiltrated water to flow laterally down the hillslope and influences the isotopic signature uniformly in a specific depth.



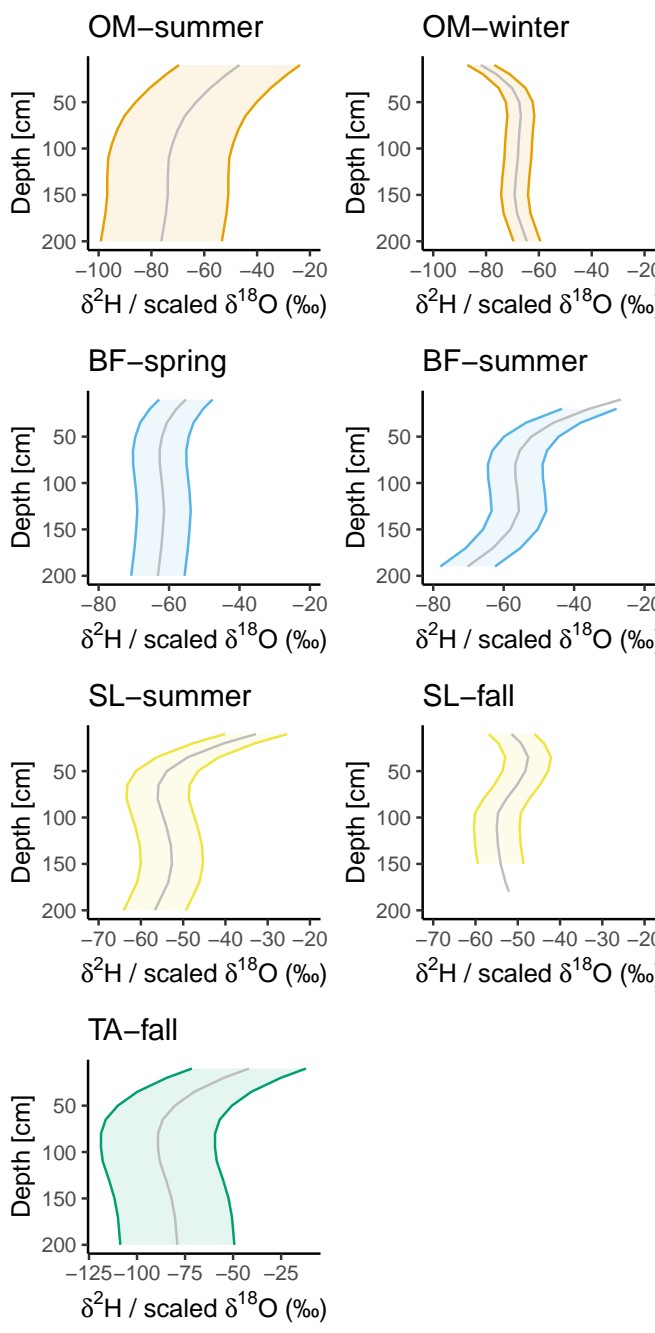

**Figure 3.** The reference profiles of Ore Mountains (OM), Black Forest (BF), Sauerland (SL) and Tyrolean Alps (TA) and their respective sampling season generated with the selected seasonality cluster and LOESS regression




**Table 1.** Summary of profile count with (multiple) preferential flow (PF) indications across Sample Groups

| Group | No. of Samples | No. of Profiles | Profiles with PF | Profiles with multiple PF |
|---|---|---|---|---|
| OM-summer | 532 | 59 | 2 | 1 |
| OM-winter | 266 | 27 | 6 | 1 |
| BF-spring | 645 | 65 | 2 | 1 |
| BF-summer | 285 | 26 | 17 | 9 |
| TA-fall | 934 | 96 | 38 | 21 |
| SL-summer | 577 | 60 | 2 | |
| SL-fall | 410 | 46 | 7 | |

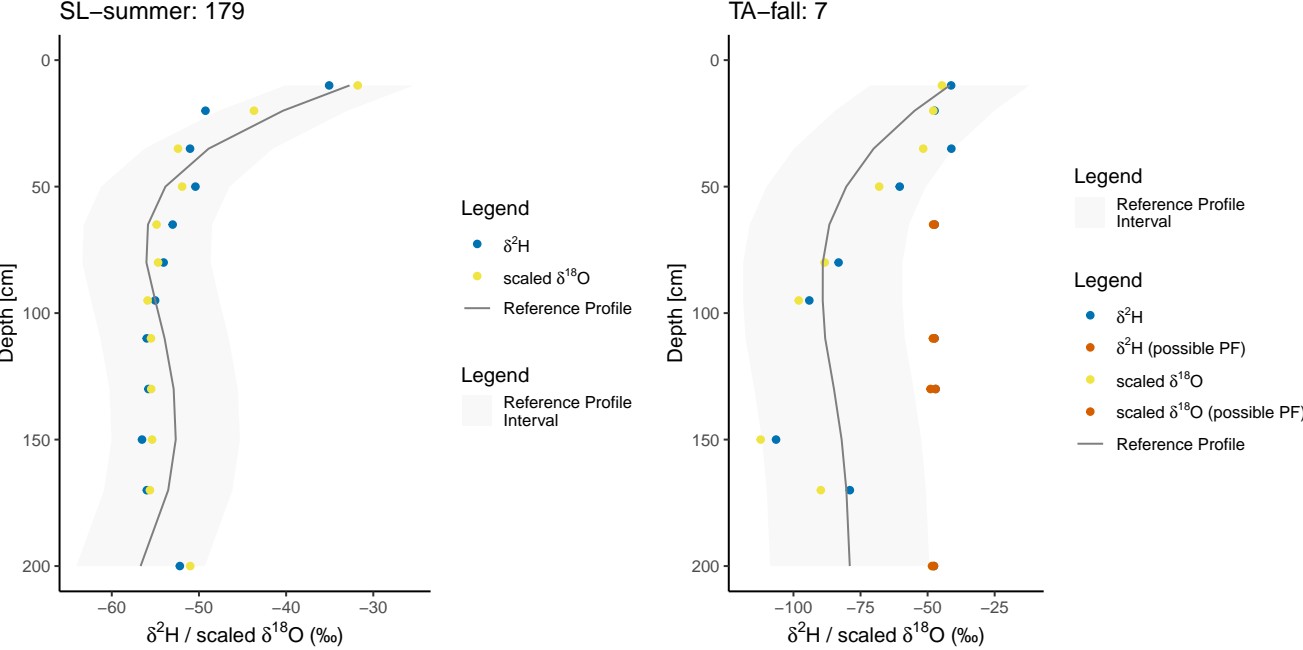

**Figure 4.** Profile Sauerland 179, where most points are within the reference profile and Profile Tyrolian Alps 7, where multiple samples (vermilion dots) diverge from the reference profile both in the measured/scaled $\delta$-values and were therefore classified as preferential flow. Yellow and blue dots show the measured/scaled $\delta$-value in each depth. The grey line portrays the generated reference profile with a $\pm$ two standard deviation band as a preferential flow identification threshold.

## 3.3 Mixing model

We applied mixing models to the isotopic compositions of the identified preferential flow samples to elucidate the composition of preferential flow. As the selected end-members were the topsoil signature and the reference signature at a given depth, a 50% result indicates an equal mixture of both "new" event water and the older, stored pre-event water. We observed no significant




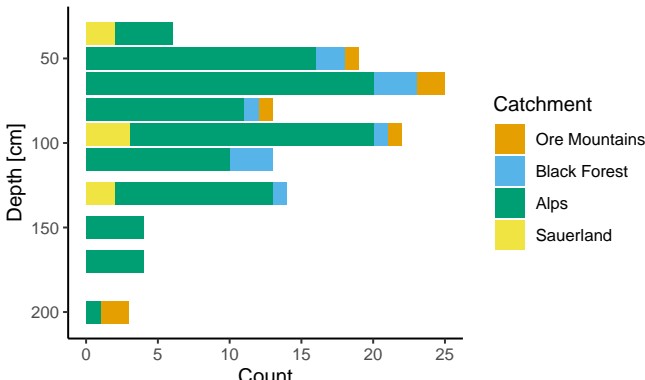

**Figure 5.** Depths in which preferential flow was detected within each catchment

differences in the mixing proportions among the catchments or across different sampling times (Kruskal-Wallis test p-value
= 0.2539). Most samples identified as preferential flow exhibited a mixture of 65% new event water and 35% older pre-event
water. There were some outliers, with values exceeding 75% or falling below 50% topsoil water (Figure 7).

### 3.4 Spatial analysis

In terms of land use, catchments with similar proportions of grassland and forest areas had an equal distribution of preferential
flow (Figure 8). In the catchments dominated by forests, specifically the Black Forest and Ore Mountains, preferential flow was
observed exclusively within the forested areas, though the count was small (n <= 7). There was a minor but non-significant trend
(Wilcox p-value = 0.06) indicating forested hillslopes in the Tyrolean Alps catchment had shallower preferential flowpaths than
grassland profiles.

Logistic regression analysis was used to assess the effect of different topographic factors on the occurrence of preferential
flow. Of the variables tested, only Aspectcos, which represents the northerly direction of slopes, showed a statistically signifi-
cant trend (p = 0.044), indicating that north-facing slopes were more likely to exhibit preferential flow than their south-facing
counterparts (Fig. 9 c). Other variables, including catchment, land use, slope and altitude, did not show significant effects.

## 4 Discussion

The developed method allowed us to identify numerous isotopic deviations from the reference profile, which we interpreted as
preferential flowpaths in the soil profiles. Scaling the $\delta^{18}$O values to the $\delta^2$H range and creating individual reference profiles for
each sampling campaign was effective for this purpose. However, the method is subject to certain assumptions and trade-offs
which we will now discuss.

One disadvantage is that a sufficient number of samples are required , with more samples improving analysis quality.
BF.summer had the fewest profiles used in a sample group, with 26; Alps.fall had the most, with 96 (Table 1). Taking more




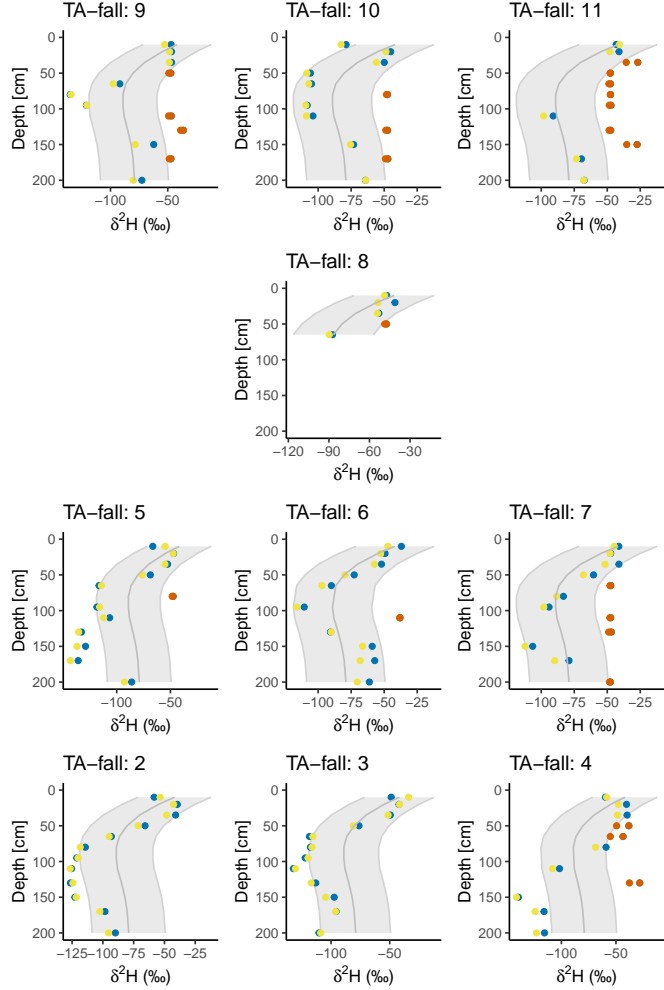

**Figure 6.** Profiles of plots 2-11 which are in close vicinity. Profile rows are spaced 10 m apart, columns 5 m. Yellow and blue dots show the measured/scaled $\delta$-value in each depth, vermilion dots indicate divergence from the reference profile. The grey line portrays the generated reference profile with a $\pm$ two standard deviation band as a preferential flow identification threshold.

samples may allow for robuster and clearer reference profiles. These improved reference profiles help to reduce uncertainty,
resulting in a more solid basis for comparison. The reduced uncertainty improves the accuracy of identifying outliers, which are classified as occurences of preferential flow. Increasing the number of samples would take more time, but it is an efficient method for achieving a catchment-wide estimation of preferential flow compared to other approaches. Dye tracer sprinkling experiments, trenches, and soil moisture sensors, while effective on the plot scale, require intensive excavation and high material and maintenance costs for permanently installed equipment. In contrast, soil core drill sampling provides a larger spatial
distribution of information at a relatively lower cost, making it a more practical solution.




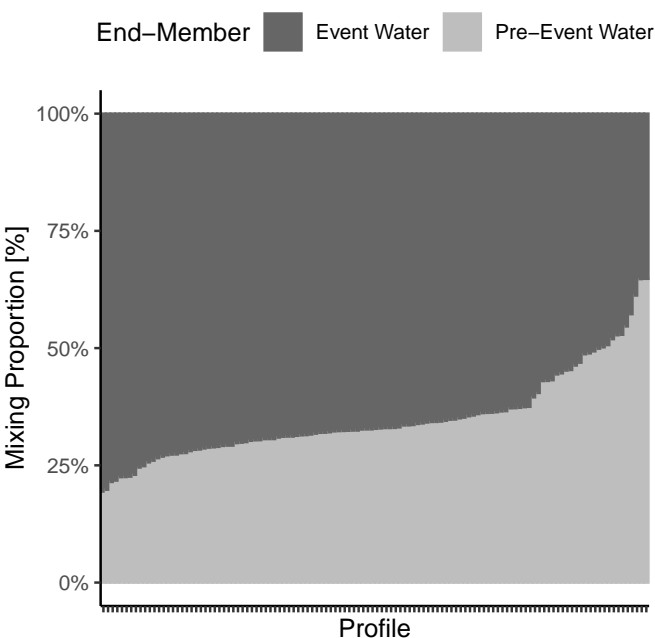

**Figure 7.** Distribution of Mixing Proportions for Preferential Flow

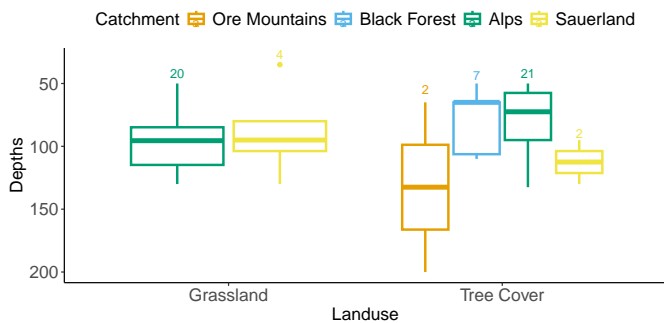

**Figure 8.** Landuse and depth of the identified profiles. Numbers above the boxplots indicate the number of observation in each.

The presented method assumes that preferential flow signatures are similar to those of topsoil. This is very likely as many tracer sprinkling experiments (e.g. Weiler and Naef (2003); Alaoui and Helbling (2006); Kodešová et al. (2012)) have shown that preferential flow consists of new infiltrated water that also seeped into the topsoil matrix. However, this is not always the case. . Preferential flow composition may differ from the topsoil signatures. For example, macropores can store older water that

is remobilized and transported preferentially during infiltration events (Klaus et al., 2013; Beven and Germann, 2013). Also topsoil hydrophobicity due to dryness or soil freezing (Täumer et al., 2006; Kelln et al., 2007; Gimbel et al., 2016; Demand et al., 2019b) funnels water into flowpaths without seeping into the topsoil matrix. In both case, the signatures of preferential




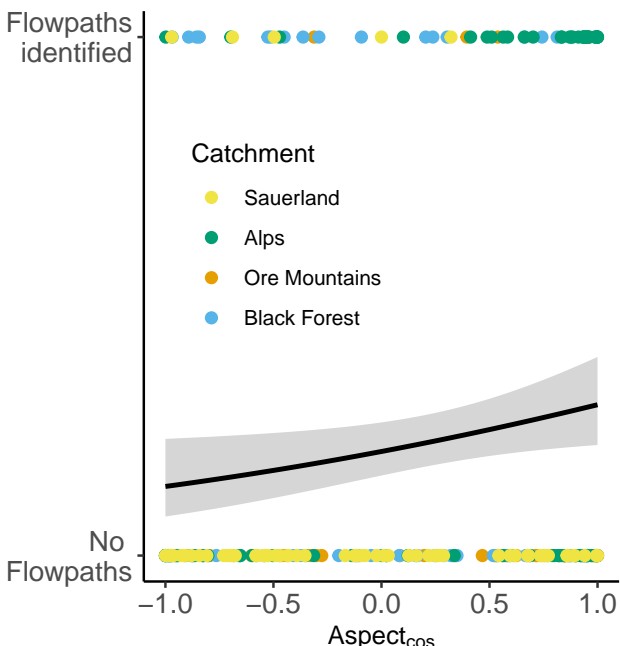

**Figure 9.** Logistic regression of the significant predictor for preferential flow occurrence Aspect$_{cos}$ (Equation 2) which distinguishes slope orientation, where -1 indicates south-facing and 1 indicates north-facing slopes

flow do not necessarily match those of the topsoil, and thuzs are not classified as preferential flow. Because sampling was not conducted during soil freezing condition and all soil samples were sufficiently moist for isotope analysis, the hydrophobicity
effect can be dismissed.

The method allows us to also observe the points where the preferential flow interacted with the soil matrix. A complex, interconnected preferential flowpath system may transport water through soils without interacting with the matrix (Anderson et al., 2009; Angermann et al., 2017). The signature of preferential flow would then be undetectable in the soil profile, as the water rushed through and out of the profile. When there is no interaction of preferential flow with the matrix, the method can
not identify these flowpaths. As a result, if the preferential flow does not interact with the matrix, the method will not detect it, and the overall occurrence of preferential flow may be higher than our estimates. However, studies that reported the missing interaction found this only in a few cases, while identifying pore-matrix interactions was typical in most cases (Anderson et al., 2009; Angermann et al., 2017).

Aside from preferential flow, sampling and analysis errors can also affect the soil water isotope profile. Because the sampling
cores are only one meter long, they must be removed after each meter of drilling. As a result, the borehole remained empty at depths of about 1 m and 2 m for a short period during sampling. During this time, water-containing soil aggregates from the overlying soil layers could have may have fallento the bottom of the borehole, altering the isotopic signature at those depths. To address this, we removed the top 5 cm of each soil core and excluded all identified flowpaths from our estimates where



the sample matched the drilling cutoff. To reduce the impact of errors during sample analysis, we scaled the $\delta^{18}$O values to the $\delta^2$H range and only considered a sample as correct if both values exceeded the reference profile. This proved to be highly effective, as 103 profiles diverged in one isotope from the reference profile while the other was within the uncertainty band.

In order to effectively differentiate between preferential flowing water and slowly moving matrix water, the catchment must have measurable isotopic seasonality. For example, the alpine catchment exhibits much larger isotopic seasonality in precipitation (range 10 ‰ in $\delta^{18}$O and 100 ‰ in $\delta^2$H) compared to the other catchments (e.g. EG range 6 ‰ in $\delta^{18}$O and 60 ‰ in $\delta^2$H)(Nelson et al., 2021). This greater seasonal contrast in isotopic composition improves the ability to distinguish between different isotopic seasons. As a result, the pronounced isotopic variations in the alpine catchment provide clearer and more distinct signals, making it easier to identify and distinguish between preferential flowing water and slowly flowing water in the soil matrix. This level of differentiation is less noticeable in other catchments with lower isotopic seasonalities, where overlapping isotopic seasons can obscure the distinction between the two flow pathways or the two water ages.

Furthermore, the alpine catchment's soil properties may be better suited to preserve seasonal isotopic signals. In contrast, the other catchments showed mostly uniform profiles, indicating extensive mixing processes like lateral flow or advective and dispersive transport that obscure the seasonal signals (Małoszewski et al., 2006; Thomas et al., 2013). According to Garvelmann et al. (2012), upslope locations exhibit isotopic seasonality in their soil profiles, while further downslope, this signal is weakened due to mixing from lateral subsurface flow. Furthermore, the influence of well mixed groundwater with typically higher water ages, which increases at the base of the slopes in the riparian zone (Uchida et al., 2005), erases the seasonality signal from the soils. Groundwater influence was also visible in the soil cores of the lower mountain ranges through redoximorphic patterns that were not present in the alpine catchment. This extensive mixing complicates the identification of the "real" seasonality profile in those regions, making it impossible to identify preferential flow clearly.

Validating the findings is difficult, as there is currently no comparable data available for quantitatively identifying preferential flow pathways in the soils of these catchments. However, subsurface flow trenches installed on selected hillslopes provided evidence of the presence of preferential flow paths within these catchments (Pyschik et al., 2025b). The results can also be placed in a broader context and compared to similar studies. The catchments showed a decrease in preferential flow with depth (Fig. 5). Several other studies which investigated preferential flow (Thomas et al., 2013; Lu et al., 2022; Guan et al., 2023) found a general decrease with depth. These findings support the assumption that the results presented here follow a plausible pattern.

The mixing models used make several assumptions and are subject to uncertainty. The reference profile represents the average value of the stored pre-event matrix water in the soil, rather than the real, local value, which is difficult to determine. The topsoil signature, on the other hand, is a true measured value, albeit subject to measurement uncertainties. These measurements were performed either manually or with VapAuSa, with the following uncertainties: VapAuSa $\delta^2$H = ± 4.5 ‰, $\delta^{18}$O = ± 0.58 ‰, and manual $\delta^2$H = ± 5.7 ‰, $\delta^{18}$O = ± 0.37 ‰. As a result, the mixing proportions are only estimates and can vary up to 20% depending on the end member ranges. However, the end members of the reference profile and the topsoil are spaced farther apart than the measurement uncertainties, thus eliminating the possibility of no mixing.



The mixing model results revealed significant mixing at the depths where preferential flow was identified. This is consistent with previous research, which suggests that preferential flow is a mix of older, stored water and new event water. After a rain event, preferential flow interacts with the soil matrix (Angermann et al., 2017), resulting in a mixed isotopic signature (Leaney et al., 1993; Gazis and Feng, 2004). Although we did not sample during rain events, frequent rainfall in the preceding days or hours supports the observed mixing.

We found no relationships between land use and the overall occurrence of preferential flow; however, we did observe a preference for shallower preferential flow in forested areas within the alpine catchment. Other studies have also identified land use-specific depth variations, but their findings were reversed: shallower preferential flow in grasslands and deeper pathways in forests (Cheng et al., 2018). Other researcher indicated that land use significantly impacts the distribution and type of preferential flow paths, leading to varied discharge responses (Bachmair and Weiler, 2012; ?; Orlowski et al., 2015). It was concluded that shallower preferential flow paths lead to quicker discharge generation, whereas deeper pathways create a sponge-like behavior, quickly retaining water and then releasing it slowly (Cheng et al., 2018).

Understanding the spatial and temporal variability of the isotopic composition of precipitation could aid in the analysis. The alpine catchment with the highest elevation gradient showed the largest uncertainty in the reference profiles ($\pm$ 25 ‰) whereas Sauerland with a low gradient only had an uncertainty of $\pm$ 5 ‰. In catchments with a high elevation gradient, distributed isotopic precipitation data would allow for the quantification of the elevation effect. As this effect locally alters the temporal variability of the isotopic composition, scaling all profiles to one elevation could reduce the uncertainty of the reference profiles, allowing for higher precision to detect preferential flow. Furthermore, isotope precipitation data would allow for the construction of reference profiles using soil hydrological models. They could be used to derive an idealized reference pattern, assuming matrix flow is the only process involved (Sprenger et al., 2015). This model-based approach could potentially provide a clearer comparison against observed profiles.

# 5 Conclusions

The methodology used in this study was effective in detecting isotopic deviations that can be attributed to preferred flow processes. Notably, the alpine catchment exhibited the highest number of identifiable profiles. This is most likely due to the region's greater seasonality of stable isotopes in precipitation, which makes it easier to detect isotopic divergences. However, the analysis found no significant inter- or intra-catchment correlations with most topography variables or land use. The only significant influence on the occurrence of preferential flow was the aspect of hillslopes, with north-facing slopes having more preferential flow identified than other slopes. The study focused on soils in hillslopes, neglecting riparian zones, where lateral flow can induce mixing, affecting isotopic signatures. It would be valuable to extend this investigation to riparian or into flat areas, where vertical flow predominates, to understand how it might alter isotopic signatures differently. Additionally, incorporating other environmental tracers such as environmental DNA (eDNA) or dissolved organic carbon (DOC) in future studies could enhance the robustness and resolution of the analysis. This multidisciplinary approach could provide deeper insights into the dynamics of preferential flow and their underlying mechanisms.



*Author contributions.* Both authors designed the experiment. JP collected and analyzed the samples, conducted the data analysis and wrote the first draft. Both authors revised and approved the final manuscript. MW secured the funding for the study

*Competing interests.* At least one of the (co-)authors is a member of the editorial board of Hydrology and Earth System Sciences.

*Acknowledgements.* We want to thank all the technicians and people involved in fieldwork without whom it would have been impossible to gather all the soil samples (Julain Reichstein, Lars Morgenroth, Sören Köhler, Yvonne Schadewell, Benjamin Gralher, Florenz König, Jürgen Strub, Tamara Leins, Nils Jansen, Annika Feld-Golinski and Peter Chifflard); We also want to thank Tamara Leins for proofreading. This research has been supported by the Deutsche Forschungsgemeinschaft (project no. 453746323) through the research unit FOR 5288: "Fast and Invisible: Conquering Subsurface Stormflow through an Interdisciplinary Multi-Site Approach". ;



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
