# Peer review of "Detecting the occurrence of preferential flow in soils with stable water isotopes"

_EGUsphere, 2025_

## Author Comment (AC2)

**Authors' responses to the comments of Ryan Stewart**

We appreciate your review and comments on our manuscript, "Detecting the occurrence of preferential flow in soils with stable water isotopes". Your feedback is valuable to us, and we will make the recommended revisions accordingly. We provide detailed responses to each of your comments below.

General comments:

This manuscript describes the results of using stable water isotopes extracted from different soil depths to detect incidence of preferential flow, using data collected seasonally from four catchments in Europe. The manuscript is well-written save for some editorial suggestions listed below, and the approach is an interesting contribution. The manuscript focuses a bit more on the validation and assumptions of the approach than on the scientific contribution itself, so having a better balance with the actually scientific contribution of the work would be something to consider during the revision process.

In terms of results and interpretations, the approach appears to work best in areas with distinct seasonality in the isotopic signature of precipitation (e.g., the Alpine catchment) and operates under the assumption that water moving via preferential flow has the same isotopic signature as the topsoil. This assumption does not cover all instances of preferential flow, as discussed starting around Line 310.

The approach also relies on the assumption that there is isotopic exchange/equilibrium between mobile and bulk matrix water (Lines 321-328). Another possibility is that mobile water could still be present (but not equilibrated) within the soil at the time of sampling. Water content at the time of sampling could be a way to distinguish between these possibilities, and it could be generally instructive to present the water content data if those are available.

Thank you for this suggestion. We will further discuss this in the revision, however we do not have water content data.

Another assumption (which I don't think was discussed) is that the reference profiles were not affected by preferential flow. The authors constrain what they consider to be reference profiles based on presumed seasonality in the isotopic signature of input precipitation, but it might also be possible to use a simple plug flow calculation or something similar to perform a back-of-the-envelope verification of the approximate depths those seasonal inputs would move given an assumption of effective porosity and rainfall depth (the latter perhaps minus ET).

Unfortunately, we lack the precipitation data of the year previous to drilling at those locations. Our approach aimed at filling this data gap with the reference profiles. We will make this clearer in the revision, that the reference profile creation was due to lacking isotope values in the rainfall.

The manuscript also does not discuss situations in which the water at depth has a depleted isotopic signature relative to the reference profiles, even though such scenarios 1) exist in the dataset, and 2) may reveal interesting behaviors about subsurface flow.

Yes, those instances are sometimes found in the data, however this was a really minor case. It might be that "older" winter water was transported there, but since we are not able to assess the origin of the signature (as it is not within the topsoil reference profile), we are unable to say which flow process transported it there.

With some revision the paper is a good candidate for publication in HESS.

Specific comments:

Line 2-16: Some minor grammatical issues here: 1) "may be quickly activated" sounds like it is referring to the soil matrix, 2) "and enhance infiltration or interflow"; 3) "profiles of stable water isotopes"; 4) "and selected those that matched…" 5) "heterogenous soils, many profiles"; 6) "flow pathways and highlight the…"; 7) "hillslope and catchment scales".

We will fix that

Line 19: "water flows through… rather than the surrounding…"

We will change this

Line 22: This range of preferential flow (16-27%) seems very specific. Either provide the context or consider not putting these numbers.

It was stated in the publication but is indeed very specific, therefore we will remove the numbers in the revision

Line 25: "understanding" may not be needed twice.

We will rephrase

Line 38: "soil-layer, which".

We will change that

Line 46: "formed".

We will update the sentence

Line 47: Check your references that "Glenn V." is needed for Wilson.

We will revise the citation to leave out the surname.

Line 53: 190 m is another number that seems very specific and context-specific. Can you at least express on a relative basis or something? Otherwise it doesn't seem transferrable.

It was also stated in that publication but is also very specific, therefore we will remove the number in the revision

Line 58: Why is SSF only limited to 1/3 of events? Another specific rather than universal number.

We will also remove this number

Line 60: Velocities have already been discussed in L27-28.

We will remove this in the revision

Line 61-65: This section could be written more concisely.

We will try to make this more concise.

Line 85: GPR can also be used in a timelapse manner to visualize changes in water content during discrete flow events.

Thank you, we will add that

Line 102: no dash is needed between continent and effect.

We will remove this

Line 111: "making them an ideal".

We will update the wording

Line 130: "and reach deeper".

We will change this

Line 132: "the signatures shift".

We will revise this

Line 140: This could be a place to elaborate on the validity/assumptions of deriving reference profiles from the dataset itself.

We will include a few sentences in the revisions where we elaborate on the validity

Line 163: add "for delta18O" after 1.7 per mille. Same for Line 166.

Yes, we missed that and will add it

Line 216: It seems you could solve this equation explicitly for f, which may not be that important but would better illustrate the analysis.

True, we will change it so it gets clearer what we are actually solving

Line 272: Interesting interpretation/context.

Thank you!

Figure 4: The shading for SD is faint (didn't show up in the printed copy I made) and would benefit from lines as in Figure 6.

We will change it to the same style as in Figure 6

Line 285: Might indicate that 65 and 35% are approximate values.

Yes, we will add that

Line 284: Suggest calling the variable "aspect", not "Aspectcos", and it indicates the direction (not just northerly).

We also calculated this for Aspect (Range 0-360 degree) which discerns between east and west facing slopes, where we found no effect, and used the cos of aspect to gain a metric to discern between north and south, therefore we will keep the name but also add some explanation on this

Line 305: Seems like these two sentences could be combined. Somewhat redundant currently.

We will write this more concisely

Line 317: "in both cases."

We will change the wording

Figure 9: Did you consider/try fitting logistic models to the individual sites? The response appears to be driven primarily by the Alps site, which makes sense as the one showing the most preferential flow, but also it is hard to see the different catchments with the points overlaid on each other.

Yes, we tried and still saw the effect, but stronger in the alps. We will add that to the revision

Line 318: "thus".

We will change this

Line 334-336: The comparison of the two stable isotopes is an important point, but one that wasn't emphasized until here near the end of the paper. It would be good to make this point early on.

We will shift this further up

Line 354-360: This paragraph is choppy in its structure and logic. I suggest rewriting.

We will rewrite it ensure more structure and logic.

---

## Author Response (AR1)

**General changes made**

- We added more detail to methods, results and discussions, explicitly stating the limits of the methods
- We added more pro/con to the conclusion
- We fixed all the grammatical and typing errors

**Authors' responses to the comments of Ryan Stewart**

We appreciate your review and comments on our manuscript, "Detecting the occurrence of preferential flow in soils with stable water isotopes". Your feedback is valuable to us, and we will make the recommended revisions accordingly. We provide detailed responses to each of your comments below.

General comments:

This manuscript describes the results of using stable water isotopes extracted from different soil depths to detect incidence of preferential flow, using data collected seasonally from four catchments in Europe. The manuscript is well-written save for some editorial suggestions listed below, and the approach is an interesting contribution. The manuscript focuses a bit more on the validation and assumptions of the approach than on the scientific contribution itself, so having a better balance with the actually scientific contribution of the work would be something to consider during the revision process.

In terms of results and interpretations, the approach appears to work best in areas with distinct seasonality in the isotopic signature of precipitation (e.g., the Alpine catchment) and operates under the assumption that water moving via preferential flow has the same isotopic signature as the topsoil. This assumption does not cover all instances of preferential flow, as discussed starting around Line 310.

The approach also relies on the assumption that there is isotopic exchange/equilibrium between mobile and bulk matrix water (Lines 321-328). Another possibility is that mobile water could still be present (but not equilibrated) within the soil at the time of sampling. Water content at the time of sampling could be a way to distinguish between these possibilities, and it could be generally instructive to present the water content data if those are available.

Thank you for this suggestion. We will further discuss this in the revision, however we do not have water content data.

Upon further evaluation, we think that when the un-equilibrated, mobile water in a depth does not contain the signature which the reference profile associates with that depth, then, no matter the water content, preferential flow must have brought the water to that depth and such an outlier would be classified as such.

Another assumption (which I don't think was discussed) is that the reference profiles were not affected by preferential flow. The authors constrain what they consider to be reference

profiles based on presumed seasonality in the isotopic signature of input precipitation, but it might also be possible to use a simple plug flow calculation or something similar to perform a back-of-the-envelope verification of the approximate depths those seasonal inputs would move given an assumption of effective porosity and rainfall depth (the latter perhaps minus ET).

Unfortunately, we lack the precipitation data of the year previous to drilling at those locations. Our approach aimed at filling this data gap with the reference profiles. We will make this clearer in the revision, that the reference profile creation was due to lacking isotope values in the rainfall.

We stated this more clearly in the method section

The manuscript also does not discuss situations in which the water at depth has a depleted isotopic signature relative to the reference profiles, even though such scenarios 1) exist in the dataset, and 2) may reveal interesting behaviors about subsurface flow.

Yes, those instances are sometimes found in the data, however this was a really minor case. It might be that "older" winter water was transported there, but since we are not able to assess the origin of the signature (as it is not within the topsoil reference profile), we are unable to say which flow process transported it there.

We added a sentence on this to the methods

With some revision the paper is a good candidate for publication in HESS.

Specific comments:

Line 2-16: Some minor grammatical issues here: 1) "may be quickly activated" sounds like it is referring to the soil matrix, 2) "and enhance infiltration or interflow"; 3) "profiles of stable water isotopes"; 4) "and selected those that matched…" 5) "heterogenous soils, many profiles"; 6) "flow pathways and highlight the…"; 7) "hillslope and catchment scales".

We will fix that

We fixed the errors

Line 19: "water flows through… rather than the surrounding…"

We will change this

Changed the wording

Line 22: This range of preferential flow (16-27%) seems very specific. Either provide the context or consider not putting these numbers.

It was stated in the publication but is indeed very specific, therefore we will remove the numbers in the revision

Removed the numbering

Line 25: "understanding" may not be needed twice.

We will rephrase

Rephrased the sentence

Line 38: "soil-layer, which".

We will change that

Fixed this

Line 46: "formed".

We will update the sentence

Corrected the spelling

Line 47: Check your references that "Glenn V." is needed for Wilson.

We will revise the citation to leave out the surname.

We fixed the citation.

Line 53: 190 m is another number that seems very specific and context-specific. Can you at least express on a relative basis or something? Otherwise it doesn't seem transferrable.

It was also stated in that publication but is also very specific, therefore we will remove the number in the revision

Rephrased the sentence so it does not mention numbers but still explain the concept

Line 58: Why is SSF only limited to 1/3 of events? Another specific rather than universal number.

We will also remove this number

We removed the whole sentence

Line 60: Velocities have already been discussed in L27-28.

We will remove this in the revision

We kept it in here because this sentence is about lateral preferential flow as SSF while the first mentioning was about Preferential flow in general

Line 61-65: This section could be written more concisely.

We will try to make this more concise.

We rewrote the section

Line 85: GPR can also be used in a timelapse manner to visualize changes in water content during discrete flow events.

Thank you, we will add that

Added this to the section referencing Angermann et al. 2017

Line 102: no dash is needed between continent and effect.

We will remove this

Changed this

Line 111: "making them an ideal".

We will update the wording

Corrected the wording

Line 130: "and reach deeper".

We will change this

Changed the spelling

Line 132: "the signatures shift".

We will revise this

Revised the sentence

Line 140: This could be a place to elaborate on the validity/assumptions of deriving reference profiles from the dataset itself.

We will include a few sentences in the revisions where we elaborate on the validity

We added this to the methods

Line 163: add "for delta18O" after 1.7 per mille. Same for Line 166.

Yes, we missed that and will add it

Added the delta 18O signs

Line 216: It seems you could solve this equation explicitly for f, which may not be that important but would better illustrate the analysis.

True, we will change it so it gets clearer what we are actually solving

We transformed the equation

Line 272: Interesting interpretation/context.

Thank you!

Figure 4: The shading for SD is faint (didn't show up in the printed copy I made) and would benefit from lines as in Figure 6.

We will change it to the same style as in Figure 6

We changed the style to that of Figure 6

Line 285: Might indicate that 65 and 35% are approximate values.

Yes, we will add that

Indicated this in the Text

Line 284: Suggest calling the variable "aspect", not "Aspectcos", and it indicates the direction (not just northerly).

We also calculated this for Aspect (Range 0-360 degree) which discerns between east and west facing slopes, where we found no effect, and used the cos of aspect to gain a metric to discern between north and south, therefore we will keep the name but also add some explanation on this

We correctly formatted that variable and linked to equation 2 here so the purpose of cosine transformation gets clearer

Line 305: Seems like these two sentences could be combined. Somewhat redundant currently.

We will write this more concisely

Summarized the two sentences in one

Line 317: "in both cases."

We will change the wording

Changed the wording

Figure 9: Did you consider/try fitting logistic models to the individual sites? The response appears to be driven primarily by the Alps site, which makes sense as the one showing the most preferential flow, but also it is hard to see the different catchments with the points overlaid on each other.

Yes, we tried and still saw some effect, but stronger in the alps. We will add that to the revision

We added a more detailed view in the revision

Line 318: "thus".

We will change this

Corrected

Line 334-336: The comparison of the two stable isotopes is an important point, but one that wasn't emphasized until here near the end of the paper. It would be good to make this point early on.

We will shift this further up

We strengthened this point in the methods section

Line 354-360: This paragraph is choppy in its structure and logic. I suggest rewriting.

We will rewrite it ensure more structure and logic.

We rewrote this paragraph

**Authors' responses to the comments of Inge Wiekenkamp**

We appreciate your review and comments on our manuscript, "Detecting the occurrence of preferential flow in soils with stable water isotopes". Your feedback is valuable to us, and we will make the recommended revisions accordingly. We provide detailed responses to each of your comments below.

I have read the manuscript entitled "Detecting the occurrence of preferential flow in soils with stable water isotopes" written by Jonas Pyschik and Markus Weiler. In this work, they introduce a new method to identify locations where preferential flow occurs (both lateral and vertical), based on stable water isotope profiles in the soil. The core idea is to use deviations from reference profiles in the isotope signature as an indication of preferential flow occurrence.

I am confident that this paper is highly relevant and fits well within the scope of HESS, as it introduces a new method to identify preferential flow using stable water isotopes in soil profiles. It highlights the potential of isotopes as a novel tool to trace subsurface processes. Although the scientific method is clearly described, the assumptions and limitations could be discussed in more detail (see, for example, general comment B).

The paper is well-structured and demonstrates the authors' strong expertise in both hydrological processes and isotope applications, here specifically focusing on subsurface flow. Figures are informative and visually appealing, and the dataset compiled is impressively broad. With some targeted refinements, the manuscript can be further strengthened and made even more accessible to a broader readership.

Thank you!

**General comments:**

**A) Reference profile**: I really like your approach, but was wondering about the following limitation. In your clustering and reference profile determination, you assume that no preferential flow is present. I can imagine that this is not always the case, because there might be particular features in some of these soils that can always facilitate preferential flow. In a way this approach means that more permanent features (facilitating preferential flow) that could be present in a lot of profiles are not indicated and present in this case. However, I can understand also that based on this average cluster profile that you create, in a way, such feature would need to be present in all the clustered profiles and at the same depth (which might be unlikely if you include a lot of different profiles and use a small number of clusters). Perhaps one could elaborate on this.

This is partially correct and we can further elaborate on this in the revision. There may be preferential flow features persistent in many profiles. However, our first step in defining reference profiles is fitting a 3rd order polynomial function through the points, which discards single outliers which are typical for preferential flow effects. However, if for instance all hillslope would laterally transports water preferentially and similar depths are influenced, then even the polynomial lines would be skewed. However, we consider this

highly unlikely. But we can certainly provide some more background on our assumptions.

We added the reason for choosing a polynomial first to smooth out the influences of PF to the method section.

**B) Discussion (1):** While the discussion already touches on important opportunities and limitations, I believe it could benefit from further elaboration on a few points:

(1) Are there particular profiles, landscape types, or environmental conditions where this method is less applicable or might fail?

We will add something on that in the revision

We added a sentence on that in the discussion

(2) Although the method is insightful, it still requires soil sampling, meaning that it provides a snapshot in time. How representative is this snapshot for a full season or hydrological cycle?

True, it is only a snapshot. We will elaborate on this in the revision. As preferential flow features are not always active, this can definitely only be seen as a short time assessment. However, some features, like the ones shown in the alpine hillslope which seem to be interconnected are probably more persistent. In future studies, multiple sampling at different times in the year could help to tackle the questions.

We added a paragraph on this in the discussion.

(3) Regarding the 2*std criterion for detecting deviations from the reference profile: could the threshold be sensitive to sampling design or the number of samples? Are the chosen limits potentially too conservative, and how might this influence the risk of false positives or false negatives?

The limits are really strict and using for instance only 1*sd we get way more preferential flowpaths identified, but also a higher probability for false positives. We will include this in the discussion.

We added more on this in the method section.

Some of these suggestions are further elaborated in the detailed comments.

**C) Discussion (2) – add more on Novelty and Contribution:** The manuscript presents an innovative and promising approach to identifying preferential flow in soils using stable water isotopes, which is a valuable addition to the "hydrological toolbox for preferential flow identification". However, it would be helpful if the authors could explicitly position their method with respect to other common approaches (e.g., dye tracer studies, hydrometric techniques, geophysical methods). This contextualization could be more clearly revisited in the discussion to clarify the unique contributions and potential complementarities of this isotope-based approach.

We will clarify the unique contributions and potential complementarities of our approach in the discussion.

We added a new paragraph on this to the discussion

**D) Accessibility and Reproducibility:** Given the method involves data clustering and isotope profile analysis, some information on data availability, reproducibility, or whether the authors will provide scripts/code for other researchers to apply this method could be valuable. Encouraging open science practices would enhance the method's uptake and usability by the community.

We will add the code to reproduce the findings. The data will be made available in a separate data paper including a wider variety of isotope data of the catchment in the following months.

We added the R code to the supplements

**E) Spatial Drivers:** The spatial analysis section appears relatively brief and less extensive compared to other parts of the manuscript. This is likely constrained by the limited number of profiles where preferential flow (PF) was detected, which reduces the ability to identify clear spatial drivers. It might be helpful for the authors to acknowledge this limitation explicitly and, if possible, consider expanding the spatial analysis or discussing potential ways to improve it in future work.

We will elaborate on the limited spatial analysis due to limited identified preferential flow profiles

We added a paragraph stating the limited spatial analysis in the results

**Detailed comments:**

1. **Citation Wilson**: For the citation of Wilson et al. (2013), the first name is included in the text and in the reference list (located in the list of authors with a G). If I see this correctly, it is even the same Wilson as the one cited for 2016 (Wilson et al., 2016). As there is no other reference to Wilson, please consider adjusting this accordingly: Glenn V. Wilson, John L. Nieber, Roy C. Sidle, and Garey A. Fox: Internal Erosion during Soil Pipeflow: State of the Science for Experimental and Numerical Analysis, Transactions of the ASABE, 56, 465–478, https://doi.org/10.13031/2013.42667, 2013.

   We will fix the citation

   Fixed the citation

2. **Citation style**: I noticed that references often end with a double ")" at the end, for example here: "(Thomas et al., 2013; Demand et al., 2019a))". Is this the style the

citations are supposed to have? If not, please consider updating this accordingly.

We will correct that error

We corrected the brackets

3. **Introduction, Lines 51–53**: "During water flow in macropores, they are typically not filled entirely with water but a water film forms along the wall if the pore, leaving the central part of the pore empty." I think the "if the pore" sounds incorrect. Should "if" be replaced by "in"?

We will change the wording

We changed the wording

4. **Introduction**: I was just curious about the connection between the following two sentences:

In Lines 21–22, authors refer to the flow in the soil:

"Preferential flow does not occur during every rainfall or infiltration event, but when it is activated, it can account for a significant proportion of annual flow through the soil, ranging from 16–27% (Eguchi and Hasegawa, 2008)."

Later, they refer to the contribution of preferential flow to discharge:

"The complex network of macropores, particularly soil pipes in hillslopes, can extend up to 190 m and can contribute up to 50% to overall discharge in streams (Jones, 2010; Wilson et al., 2016)."

I was wondering if these statements fit together or if they might mismatch. If you could also give a range for the second statement ("up to 50%"), this would maybe give a better feeling of the general importance and variability in preferential flow importance.

In line with the Review by Ryan Stewart we will fully remove these numbers.

We removed the numbers and wrote it more relative

5. **Introduction, statement on Ground Penetrating Radar (GPR)**: A period is missing at the end of the sentence. This was present for the other sentences. Also, I assume that another problem of these geophysical methods is the non-unique solution of the inversion results (imaging). This could be added and stated as a limitation of the method. Perhaps you could in this section rather refer more generally to geophysical methods that are able to characterize PF (ERT – Electrical Resistivity Tomography – is, for example, also used in multiple cases to visualize preferential flow paths), and include the limitation of non-uniqueness (underdetermined system)?

We will fix the period errors and state this limitation

We fixed the error and added ERT to the List

6. **Introduction, Lines 132–134**: "Over time, however, the signatures shifts towards those of pre-event water (Leaney et al., 1993; Gehrels et al., 1998; Kelln et al., 2007) due to lateral infiltration and exchange processes between the preferential pathways and the surrounding matrix." Please replace "shifts" with "shift".

We will fix that

Fixed the wording

7. **Figure 1**: It would be great if you could add (either in the figure or the caption) that the soil profiles are stable water isotope profiles. Perhaps the authors could also add the number of profiles to each catchment (to get a feeling for the number of points – vertical profiles – in each area). Is this always 100? Then this could also just be mentioned in the caption to give a direct feeling of the magnitude.

We will add that to the figure caption and map

Added the description and also the number of profiles

8. **Methods, 2.3, Line 197**: "from the R caret package Kuhn (2008)." Probably this should be: "from the R caret package by Kuhn, 2008" (or use the bracket version: "(Kuhn, 2008)").

We will change that

Changed the citation

9. **Figure 2**: I like the schematic representation of the method, but I think it would help if the figure had x and y axis descriptions (not ranges, just an explanation of variables – depth on the y-axis, x-axis labeling $\delta^2H$ / scaled $\delta^{18}O$ (‰)). In the second sub-figure, the line borders of x and y axes are missing. I suggest implementing these as well.

We will try to include this while still keeping the graphic concise

We added axis labels to the schematic

10. **Methods, section 2.3**: "Only depths where both $\delta^2H$ and scaled 18 O values exceeded the reference range were considered for potential preferential flow (Figure 2.4)." Should "scaled 18 O" not be replaced by $\delta^{18}O$? Also, when referring to Figure 2.4, one could state the color of the points that are considered indicative of preferential flow (orange points, right?).

We will add that and revise

We added the reference to the color, however the scaled 18O is correct at that point as those values are used in that step

11. **Methods, 2.4 Mixing models**: "By solving this equation, the proportion of preferential flow (f) at the identified depths was determined." Before this sentence, a description is provided about the proportion of preferential flow. Does this implicitly mean that one could not indicate preferential flow in the topsoil with this method? Also, how would this equation work if the delta values of the topsoil and reference are very similar (could for example happen at the second sampling depth, I imagine)?

    We always exclude the two topsoil signatures as these data ranges are what we use to identify preferential flow. We would always get 100% event water in those zones.

    We added that we exclude those depths from the mixing calculations

12. **1 Reference profiles**: In the results, the sampling periods for each region are also mentioned. This could possibly be moved to the methods section to keep more focus on the results here. I think the results are very descriptive. It would be quite nice if there was more comparison between the results. For example, to more easily compare results from different areas, one could also describe all summer results together and position the plots in Figure 3 accordingly. For instance, blue and yellow curve SL summer and BF summer curve seem to have very similar features and could be connected nicely.

    We will try to find a way to arrange the graphics this differently and also add some comparative sentences

    We shifted the sampling periods to the method section and positioned the plots in columns of season. Also we added a comparative paragraph

13. **Table 1**: First of all, it's really impressive how many samples and profiles have been measured for this effort. I would suggest adjusting "Profiles with multiple PF" to "Profiles with PF at multiple depths." I also think that the number of profiles showing PF is probably highly influenced by how you set the detection boundaries for PF. You used 2 times the standard deviation, which is a sound statistical concept, but I wonder how this connects to the process/physics – i.e., the occurrence of preferential flow in general. Keep in mind that these limits (2* std) do affect the number of locations where you will detect preferential flow heavily.

    As commented above, we will add some sentences concerning the standard deviation range chosen

    We added some sentences to the method section on the reason for choosing 2 sd

14. I also think the way the data is clustered can affect the outcome of your analysis substantially. I assume that if your clustered pool is larger and has more isotopic variability (at different depths), the chance that a particular profile pops up as PF will be smaller, as compared to a small group of clustered profiles with lower general variability. This means that the sampling locations chosen and the number of locations sampled might affect the output. It also means that for reference profiles with small standard deviations, smaller deviations from the mean might already lead

to PF detection, whereas in the case of large variability, it might not. Perhaps this should be discussed in the discussion section as something important to keep in mind.

We can add this to the discussion, but basically the most important driver is the intra-catchment variability of stable water isotopes in the soil. If the variability is low, reference profiles are narrow. If it is large, so will be the profile ranges

We added something on the need to have enough representative samples to fully capture isotopic seasonal variability

15. **Figure 4**: The difference between the two PF classes here is not clearly visible color-wise. If you want to distinguish them, it would probably be better to give them more distinct colors (currently both look reddish/orange). The grey background is not very visible and could be made a bit darker to more clearly show the region where samples are considered "piston flow."

We chose the same color for both, as only if both are outliers, they get classified

16. **Figure 6**: Although I really like the figure, I did not find a clear reference to it in the main text. I was also not sure what the authors want to say with this plot. Why are fall plots from the TA location shown? What is the reasoning behind this arrangement? It would be great if the figure could be better integrated into the manuscript. In the caption, I also miss what the figure is trying to convey. To guide the reader more, I suggest clearly stating the purpose of the figure in the caption.

We can state more clearly that the aim of this plot was to visualize profiles on a single hillslope in near vicinity. The profiles in the lowest row are at the hill-foot and each row goes 10-20m upslope. Since most profiles were drilled throughout the catchment, this gridded design is present on 3 hillslopes per catchment and allows for analyzing plot scale variability on a hillslope

We added the explanation to the figure caption

17. **Figure 7**: I generally like the figure, but I think it's unclear what is on the x-axis. One could choose to depict a numerical value here (e.g., number of PF profiles) or a more meaningful variable. In the text, you write: "Most samples identified as preferential flow exhibited a mixture of 65% new event water and 35% older pre-event water." Perhaps you could show this in the graph as well (as a dotted line, for example).

We will adapt the graphic

We updated the axis label and also added spacing between the bars

18. **Discussion, Line 314**: "However, this is not always the case." Remove the second period (".").

We will revise this

We revised this

19. **Discussion, Line 302**: "One disadvantage is that a sufficient number of samples are required , with more samples improving analysis quality." Remove the space before the comma here. Regarding the number of samples: I think it is not only about the number of samples. The representativeness of the samples also really matters, especially if you generate a reference profile. What would happen if you over-sample one location and under-sample another? How would the sampling design affect the reference profile and clusters that are built?
We will change this and also include something on the representativeness of our sampling locations and the effect of them

We added something on this in the discussion

20. **Figure 9**: I would rephrase the caption to explain more clearly the intent. Also consider not using "Aspectcos" as a term here, but rather describing it in terms of slope direction. I'm not 100% sure about the absolute magnitude of the regression slope in the graph (as it does not reach "Flowpaths identified," but rather stays close to "No Flowpaths"). I assume it relates to the probability of PF (between 0 and 1)? Does this not mean that, in general, the probability of detecting PF is lower than not detecting it—which may also be related to the fact that your sampling design includes many more "no PF" samples? How does your sampling distribution here affect your analysis?

As stated in the answer to Ryan Stewart, we will elaborate on that name. And yes, it is still more probable to find a profile without preferential flow on north facing slopes than it is to find one with one, however the chance of finding a PF Profile is highest on north facing slopes.

We rewrote this section and added the explanation

21. **Discussion Section 3.4**: I think this section could benefit from more detailed interpretation. I found it quite short and was thinking that some things could be explained (e.g., more on preferential flow from a catchment perspective). In general, I also wondered whether one could really draw strong conclusions about site-specific characteristics, given that the number of profiles where PF was detected was generally quite small. In TA, there are many detections, but in some other catchments, only 2 or 6 cases. This also relates to Figure 8 and 9.

We will further elaborate the validity of the conclusions drawn from the results and also write more on the number of profiles identified

We added the limited number of identified PF to the results and also a sentence in the discussion stating the limited occurrence

22. **Connection between Introduction (Lines 64–68), Introduction and Discussion**: In the introduction, the authors mention that one of the complications of preferential flow detection is that field observations are just a snapshot:

"Also, field surveys are only a snapshot of an environmental system, but flowpath arrangements can change completely within one year (Wessolek et al., 2009; Beven, 2019). Additionally, even when preferential flow features are successfully identified at the plot scale, these findings cannot be easily extrapolated to the whole catchment."

To what degree do you think this applies to your newly proposed method? Are the profiles you take not also a snapshot of the preferential flow pathways? What kind of temporal sampling or general sampling design would be needed to get a statistically sound idea of preferential flow occurrence at a particular site, rather than just a snapshot.

As stated in the beginning, it depends on the feature, but yes, it is definitely a snapshot, and it will always be as soil cores are highly invasive and once sampled may alter the flow mechanisms surrounding the sample location

We added a paragraph on this in the discussion

23. **Conclusion:** I think the conclusion in this case would particularly benefit from a clear summary of pros and cons of this new methods. Currently the summary might not be fully accessible to people that have not read other parts of the manuscript. I always think it is a good opportunity to do this, as this would help to also guide readers that might just look at the conclusions really quick – this is helping readers quickly grasp the method's utility and potential challenges.

We will add more pro/con to the conclusion

We added a few sentences stating the advantages and downsides of this method in the conclusion

---

## Author Response (AR2)

**Response to Reviewer #1**

Thank you for your comments, we revised them according to your specifications.

Line 84: Best to not start sentences with acronyms.

We removed the acronym to be in line with the other items
Line 133: "With preferential flow, young…"

We corrected the sentence
Line 209/210: The dash (I presume?) did not format correct.

Yes, this was an issue in the Diff file but in the original its correctly formatted
Line 225: Capitalize "Figure" (proper noun).

We changed this
Line 230: "the event" (rather than "this").

We don't know what you reference here, there is no "the event" or "this event" anywhere in those lines
Line 257/266, etc.: You don't need to redefine these acronyms. Also, Line 257 should read "In the OM, the…"

We updated the sentences to only have the abbreviation
Line 288/289: Same comment on dash formatting.

Yes, sorry, not included in the real file but changed in the new diff
Line 418: Which questions?

Changed to "this issue"
Line 419: "TA-fall figure 6" doesn't make sense.

"figure 6" should be in parentheses, changed it

Line 443: This what still offers the potential? Study? Methodology?

Changed to "they", referencing the limited identified profiles